# A Multi-Party Functional Signatures Scheme for Private Blockchain

Quan Zhou [1,*], Yulong Zheng [1], Kaijun Wei [2], Minhui Chen [2] and Zhikang Zeng [1]

1   School of Mathematics and Information Science, Guangzhou University, Guangzhou 510006, China; 2112115056@e.gzhu.edu.cn (Y.Z.)
2   School of Computer Science and Cyber Engineering, Guangzhou University, Guangzhou 510006, China
*   Correspondence: zhouqq@gzhu.edu.cn

**Abstract:** Digital signature technology is essential for ensuring the authenticity and unforgeability of transactions in a private blockchain framework. In some scenarios, transactions require verification from multiple parties, each of whom needs to authenticate different parts of the transaction. To address this issue, researchers have developed multi-party ECDSA (Elliptic Curve Digital Signature Algorithm) signature schemes. However, these schemes either need to consider the authentication of different parts of the transaction or generate an aggregated signature. This paper proposes a novel solution that combines functional signatures and multi-party ECDSA signatures to create a multi-party functional signature for private blockchains. Compared to previous constructions, the proposed scheme ensures that each part of the transaction is verified. Furthermore, when the aggregate signature of the entire transaction cannot be verified, this scheme identifies the specific part of the transaction for which the signature authentication fails instead of rejecting the entire transaction. This paper uses a smart contract to securely deploy the proposed scheme and authenticate the $f$ in functional signatures. The constructed scheme also provides security under the existential unforgeability of the ECDSA signature, even if $n-1$ parties are corrupted, assuming a total of $n$ parties. The scheme of this paper successfully conducted experiments on a personal computer, with three users taking approximately 343 ms, six users taking 552 ms, and nine users taking 791 ms.

**Keywords:** functional signatures; private blockchain; multi-signature; smart contract; ECDSA signature

## 1. Introduction

Recently, blockchain has played a significant role in various fields, including finance [1–3], IoT [4,5], and decentralized storage [6] due to its decentralization, immutability, and traceability properties. The development of blockchain in cryptocurrency has made it a modern network technology [7]. Bitcoin [8], as blockchain 1.0, has successfully deployed and applied the ECDSA (Elliptic Curve Digital Signature Algorithm scheme), gaining wider attention and gradually becoming the default signature mechanism of current mainstream blockchain platforms and projects. Some blockchains, such as Ethereum [9] as Blockchain 2.0 and Hyperledger Fabric, have been proposed to make blockchain more flexible and perform better. Smart contracts, as the core technology of Ethereum [9], can be seen as computer programs that perform all contract-related operations autonomously, with corresponding results. The validity and authenticity of the contract operations can be checked on the Ethereum account. Deploying smart contracts can realize some special operations in the blockchain system. A private blockchain is a permission-based network that limits usage to specific organizations or individuals. Unlike public blockchains, private blockchains allow only authorized users or organizations to join, and their transaction data are restricted to authorized viewing. The efficient development of any network technology must consider its network security factors [10–15], and the signature technology of cryptography can effectively solve some security problems in the blockchain.

Functional signatures, a new type of signature introduced by Boyle, Goldwasser, and Ivan [16], have some properties that standard signatures do not have. In a functional signatures scheme, there are two private keys, a private key that can sign any message belonging to the message space called the master private key, and another one that can sign information defined in the scope of the function $f$. The owner of the master private key can use the master private key to generate a functional signing private key $sk_f$ for function $f$. The owner can allow others to use the functional signing private key $sk_f$ to sign messages in the range of $f$. The above gives the functional signatures the property of fine-grained access control, and the property is beneficial for multiple users. For example, there is a document that can be noted as message $m$. The primary user with the master private key can sign the $m$. The primary user only allows other users to modify some of the information in the document. It issues some functions $f$ for other users, and others can sign the $f(m)$ to modify part of the document. Based on [16], Backes [17] proposed delegatable functional signatures and achieved the application of delegation using functional signatures. In the functional signatures scheme, the master key and the signing key $sk_f$ of function $f$ are generated by one user. Once this user is corrupted, it will cause much damage. The new scheme called decentralized multi-authority functional signatures, which supports non-monotone access structures using inner-product relations, was introduced by Okamoto and Takashima [18] to address this type of problem.

In private blockchain systems, private keys are essential for users to control their cryptocurrency and are the basis for implementing privacy protection schemes. If a malicious attacker illegally steals a private blockchain user's signature private key, they can transfer the user's cryptocurrency arbitrarily or impersonate the user to publish illegal transaction information. This can lead to severe economic loss problems [19]. Some multiple signature schemes have been proposed to solve this problem. MacKenzie [20] initially proposed a two-party signature scheme for DSA that enables two signing parties to generate a valid DSA signature for a given public key. Neither party can complete the signature alone, and no reconstruction of the private key is required during the signing process. Lindell [21] proposed a two-party ECDSA signature scheme based on the Paillier homomorphic encryption algorithm. This scheme utilizes the Paillier homomorphic property to complete the signature process and improve efficiency directly. Castagnos [22] introduced hash proof systems technology to replace the Paillier homomorphic of Lindell [21] and completed the security proof. In addition to the two-party ECDSA signature scheme, multi-party threshold ECDSA signature schemes have also been proposed. Lindell [23] designed a secure privacy multiplication protocol for extending the original two-party signature scheme into a multi-party threshold signature scheme. Doerner [24] also designed a multi-party threshold signature scheme by extending the two-party signature. Gennaro [25] designed the MtA protocol based on Paillier homomorphic encryption and proposed a new multi-party threshold ECDSA signature scheme based on this protocol. However, these schemes use a homomorphic encryption algorithm or oblivious transfer (OT) protocol used in two-party signature schemes, so they have high computational and communication overheads. The above-distributed signatures are all multiple parties signing the same message, and none can solve the problem of fine-grained authentication of partial information. For example, in a technology company, various departments may need to authenticate specific project parts before uploading them to the company's private blockchain. However, the methods noted above do not allow for this level of fine-grained authentication, as they only produce an aggregated signature. This paper uses the ECDSA scheme as a standard signature in functional signatures and constructs a system model based on functional signatures with multiple parties for private blockchain. The proposed scheme enables multiple parties to sign different parts of the message and collaborate to produce a signature for the message. It can effectively solve the example described above. It can solve some real problems about fine-grained security authentication of multiple parties. The contributions of this paper are as follows:

- This paper proposes a novel scheme that combines functional signatures and multi-party ECDSA signatures to create a multi-party functional signature for private blockchains. Compared to the previous scheme, the proposed scheme ensures that each part of the transaction is verified. Moreover, in cases where the aggregate signature of the entire transaction cannot be verified, this scheme identifies the precise portion of the transaction where signature authentication fails rather than rejecting the transaction in its entirety.
- This paper ensures the security and authenticity of function $f$ in the function signature by embedding it in a smart contract using blockchain's smart contract technology.
- This paper provides proof of the security of the constructed scheme. Under the existence of the unforgeability of ECDSA signatures, the proposed scheme is secure even if it is corrupted in $n - 1$ parties (assuming a total of $n$ parties).

## 2. Related Work

The security of network technology has always been a frontier worthy of research. Shaukat [26,27] summarized the cyber security issues regarding the rapidly evolving technologies. For the development of blockchain security, Halpin [28] proposed an introduction to security and privacy on the blockchain. As blockchain technology evolves, many researchers have recognized cryptography as essential for analyzing and addressing blockchain security concerns. Li [29] utilized puncturable signatures in cryptography to resolve the issue of long-range attacks in the blockchain's Proof of Stake protocol. Signature technology is the core of cryptography. It can effectively solve many blockchain security problems.

The use of signature technology to protect the security and privacy of the blockchain has been studied by many scholars. Zhu et al. [30] proposed an interactive incontestable signature for transactions of blockchain, which ensures that the transactions were certified by the dealer and are non-repudiable. Mercer [31] applied ring signatures to the blockchain, enabling privacy protection for the blockchain users themselves. Gong et al. [32] proposed an anonymous traceability protocol based on group signature; the protocol guaranteed the anonymity of blockchain transactions. The design of multiple signatures in blockchain can effectively solve the problem of multi-party verification and decentralization of blockchain transactions. Kogias et al. [33] proposed a collective signing scheme to improve the security and performance of Bitcoin, but it was not very efficient. Zhou et al. [34] proposed a distributed account management scheme for Bitcoin, which aims to protect the private key of a user's account. Alangot et al. [35] introduced a spanning-tree topology to scale Schnorr multi-signature, which could enhance the consistency of the blockchain. Yu [36] proposed a method to combine Schnorr multi-signature and blockchain to ensure privacy and security of data in the Internet of Things. Maxwell [37] constructed a new Schnorr multi-signature scheme, which could improve both performance and user privacy in Bitcoin. Yu [38] also proposed an elliptic curve threshold signature scheme for blockchain, which has relatively high efficiency compared to other multi-party signatures of elliptic curves. Xiao [39] proposed a secure and efficient multi-signature scheme for Fabric, and Uganya [40] proposed a modified elliptic curve cryptography multi-signature scheme to enhance security in cryptocurrency. The detailed comparison is shown in Table 1.

All of the above blockchain multi-signature schemes only generate one signature collaboratively by multiple parties, without considering the fine-grained authentication of partial transaction data by a single party before collaboratively generating a single signature. So far, no scheme has considered fine-grained authentication of transactions in the private blockchain along with multiple signatures. This paper proposes a multi-party functional signatures scheme based on ECDSA signature that can solve the aforementioned problems.

**Table 1.** Comparison of different multi-signatures.

| Scheme | Efficiency | Blockchain | Provable Security | Against Collusion Attack |
|--------|-----------|-----------|------------------|-------------------------|
| [24] | low | no | yes | no |
| [33] | low | yes | uncertain | no |
| [34] | low | yes | uncertain | yes |
| [37] | high | yes | yes | yes |
| [38] | high | yes | yes | no |
| [39] | high | yes | yes | yes |
| [40] | high | yes | yes | yes |

## 3. Preliminaries

Some basic definitions and notational concepts are as follows.

### 3.1. ECDSA Signature

ECDSA signature is divided into four steps, which include Setup, KeyGen, SigGen, and Verify. Setup is responsible for generating the system parameters, KeyGen generates the public–private key pairs of the scheme, SigGen represents the signing process, and Verify represents the verification process of the signature. The detailed steps are shown below.

*Setup:* On inputting the security parameter, the system outputs public parameters param = {E, $F_p$, G, P, p, q, H}, where E is an elliptic curve defined over a finite field $F_p$, p is a prime and G is an additive cyclic group consisting of all points in E; P is the generator of the group G, and q is the prime order of the group G. Finally, H is a cryptographic hash function denoted as $H : \{0,1\}^* \rightarrow Z_q^*$ and $Z_q^*$ is the field consisting of the set of integers $\{1, 2, \ldots, q-1\}$.

*KeyGen:* On inputting the public parameter param, the following two steps generate the signed public–private key pair.

- Choose a random integer $d \in Z_q^*$ as secret key.
- Compute the $Q = d \cdot P$ as public key.

*SigGen:* On inputting the public parameter param, the signing secret key d, and the message m, output the signature of message m $\delta = (r, s)$. The steps for generating a signature are as follows.

- Randomly select integer $k \in Z_q^*$.
- Compute $R = k \cdot P = (r_x, r_y)$, where R is a point on the elliptic curve E.
- Compute $r = r_x \bmod q$, and if/when r is 0, return to the first step to reselect k.
- Compute $s_1 = k^{-1}(e + dr) \bmod q$, where $e = H(m)$.
- Output the generated signature $\delta = (r, s)$, where $s = min\{q - s_1, s_1\}$.

*Verify:* On inputting the message m to be verified and it's signature $\delta$, output 0 or 1 (0 means failure, 1 means success). The steps for verifying a signature are as follows.

- Check whether the integers r and s belong to $Z_q^*$. If they do not belong, terminate the verification; otherwise, execute next step.
- Compute $e = H(m)$.
- Compute $R' = s^{-1}(eP + rQ) = (r_{x'}, r_{y'})$ to verify signature.
- When $r = r_{x'} \bmod q$, output 1, otherwise output 0.

### 3.2. Functional Signatures

**Definition 1.** *Functional signatures (FS) scheme for a message space $\mathcal{M}$ and a function family $\mathcal{F} = \{f : \mathcal{D}_f \rightarrow \mathcal{M}\}$; it has four algorithms as follows.*

- *FS.Setup $(1^\lambda) \rightarrow (Msk, Mvk)$ : On inputting a parameter $1^\lambda$, it can output the master signing key Msk and the master verification key Mvk.*
- *FS.KeyGen(Msk,f) $\rightarrow sk_f$ : On inputting the master signing key and a function $f \in \mathcal{F}$, it outputs the signing key $sk_f$ for $f$.*

- *FS.Sign*(f,$sk_f$,m) $\rightarrow$ $(f(m), \sigma)$ : On inputting the function $f \in \mathcal{F}$, generated signing key $sk_f$, and the message $m \in \mathcal{D}_f$ that needs to be signed, it outputs the pair signature $(f(m), \sigma)$.
- *FS.Verify*(Mvk,$m^*$,$\sigma$) $\rightarrow$ $\{0,1\}$ : On inputting the master verification key Mvk, a message $m^*$ and a signature $\sigma$, where $m^* = f(m)$, it outputs 1 or 0, where 1 indicates that the signature is valid.

### 3.3. Security Definition

**Definition 2.** *Unforgeability: The functional signatures is unforgeable if the advantage of any PPT algorithm $\mathcal{A}$ in the following game is negligible:*

- The challenger runs $(Msk, Mvk) \leftarrow$ FS.Setup($1^\lambda$) and sends Mvk to adversary $\mathcal{A}$.
- Adversary $\mathcal{A}$ is allowed to query the key generation oracle and a signing oracle; they are noted as $\mathcal{O}_k$ and $\mathcal{O}_{sig}$. The challenger initializes a dictionary indexed by tuples $(f, i) \in \mathcal{F} \times \mathcal{N}$, which contained the signing keys: $sk_f^i \leftarrow$ FS.KeyGen(Msk,f). This dictionary records the keys that were previously generated in the Unforgivability game. $\mathcal{O}_k$ and $\mathcal{O}_{sig}$ are defined as follows:
  - $\mathcal{O}_k$: On inputting $(f, i)$, the challenger runs as follows:
    1. If there is an entry for $(f, i)$ in the dictionary, then the corresponding key $sk_f^i$ is output.
    2. Otherwise, by $sk_f^i \leftarrow$ FS.KeyGen(Msk,f) sample a fresh key, add the $(f, i) \rightarrow sk_f^i$ to the dictionary in order to update it, and output $sk_f^i$.
  - $\mathcal{O}_{sig}$: On inputting $(f, i, m)$, the challenger runs as follows:
    1. If there is an entry for $(f, i, m)$ in the dictionary, then the corresponding signature $\sigma \rightarrow$ FS.Sign(f,$sk_f^i$,m) is output.
    2. Otherwise, by $sk_f^i \leftarrow$ FS.KeyGen(Msk,f) sample a fresh key, add the $(f, i) \rightarrow sk_f^i$ to the dictionary, and the corresponding signature $\sigma \rightarrow$ FS.Sign(f,$sk_f^i$,m) is output.
  - The adversary wins the game if it can output a signature $(m^*, \sigma)$ such that:
    1. FS.Verify(Mvk,$m^*$,$\sigma$) = 1.
    2. There does not exist m such that $m^* = f(m)$ for any $f$ that was sent as a query to the key generation oracle $\mathcal{O}_k$.
    3. There does not exist a $(f, m)$ that was queried to the signing oracle $\mathcal{O}_{sig}$ and $m^* = f(m)$.

### 3.4. Smart Contract

In a private blockchain system, the smart contract is compiled by the Solidity language and each different node represents a different private blockchain account. The smart contract is a type of computer protocol that can be executed through the account. Briefly, the smart contract is a computer program that has been compiled and deployed on the private blockchain. The process of deploying smart contracts is shown in Figure 1.

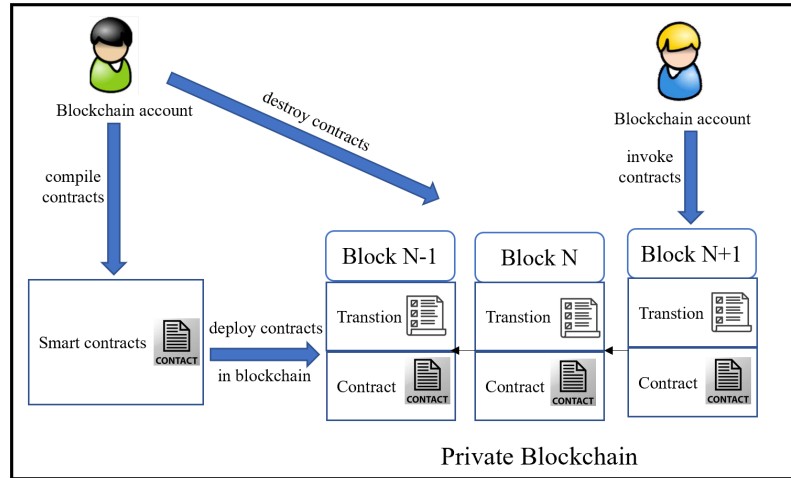

**Figure 1.** Smart contract deploy and invoke process on the private blockchain.

## 4. Proposed System Model and Multi-Party Functional Signatures Method

This section gives the concrete system architecture framework proposed in this paper. Moreover, it gives the specific setup of the scheme and the construction of smart contracts to run.

### 4.1. System Model Framework

The concrete system model is shown in Figure 2. When a transaction in the private blockchain requires fine-grained collaborative authentication by multiple nodes, the full nodes prefer to send the parameter of the signature scheme to each mining node. A complete transaction is recorded as *Data*, and a partial transaction is recorded as $Data_i$. The detailed steps are described below:

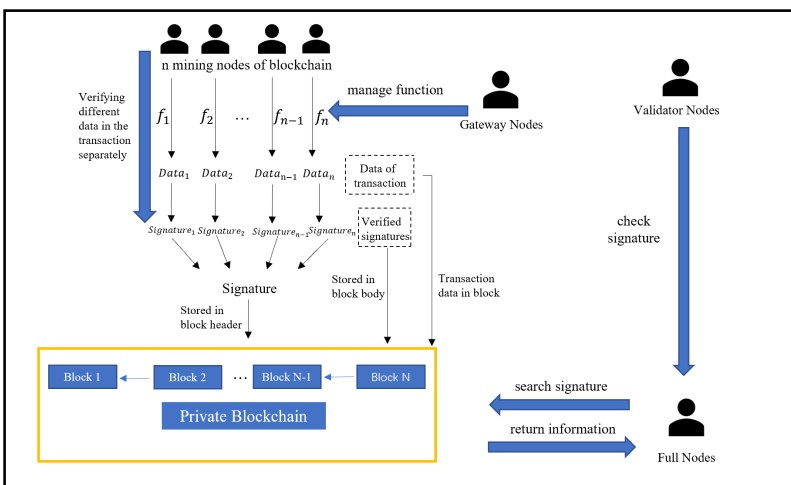

**Figure 2.** The framework of the system model.

- Gateway nodes manage the function *f* of functional signatures using smart contracts.
- Each mining node checks different $Data_i$ of *Data*, divided by function *f* of functional signatures. After that, the signature $signature_i$ corresponding to the respective $Data_i$ is generated. Gateway nodes manage the function *f* of functional signatures using smart contracts.
- After receiving all partial transaction signatures $signature_i$, an aggregated signature *Signature* is generated and stored in the block header.
- Full nodes can search for all the transaction signatures stored in the blockchain.

- Validator nodes check the correctness of the queried signature. Once an error occurs in the signature stored in the block header, the partial transaction signature $signature_i$ stored in the block body can be queried and checked.

As is shown in Figure 3, the final aggregated signature is stored in the block header, and the signatures generated for different transactions are stored in the respective transactions. The validator nodes can query the signatures of the different transactions from full nodes and verify that the signatures are valid.

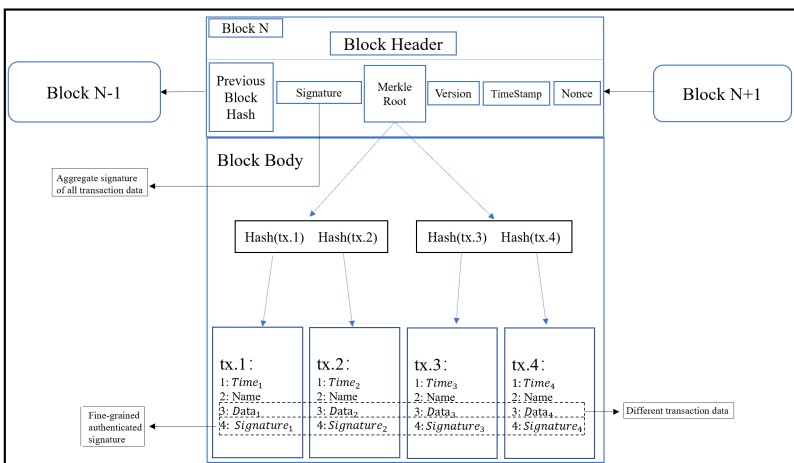

**Figure 3.** Simple block structure.

### 4.2. Smart Contract Settings and Proposed Concrete Scheme

This subsection gives the specific smart contract setup and scheme construction.

#### 4.2.1. Smart Contract Design

This section introduces the concrete process of deployment and invocation of smart contracts and related algorithms. Some special variables are used in the algorithm of smart contracts: authorizeAccount is a set of trusted private blockchain accounts (nodes of private blockchain), functionSet stores all functions $f$ of functional signatures, ct.sender is the account, and f.Set stores the account of who can use $f$. The specific definitions of these variables are as follows:

- *authorizeAccount:* When deploying the smart contract, the gateway nodes collect all legal account identity information. All these accounts consist of authorizeAccount.
- *functionSet:* The set stores all the functions $f$ of functional signatures. Functions $f$ of functional signatures that do not belong to functionSet are prohibited from being invoked.
- *ct.sender:* This invokes the contract's account.
- *f.Set:* All functions that are stored in functionSet have a set f.Set, and some accounts are stored in it. If ct.sender does not belong to f.Set, it cannot use $f$.

The concrete algorithm for the contract is as follows:

*invokeF(f):* Algorithm 1 can be executed by legitimate accounts. The gateway nodes determine whether the function $f$ invoked by this node is legitimate by the returned value.

*addUser(f,ct.sender):* Algorithm 2 shows that the system model can dynamically endow nodes with the use of function $f$.

*deleteUser(f,ct.sender):* Algorithm 3 shows that the system model can dynamically reject the node's use of the function $f$.

---

**Algorithm 1:** invokeF($f$)

---

**Input:** $f$
**Output:** bool
**if** *ct.sender does not exist in authorizeAccount* **then**
    throw;
**end**
**if** *f does not exist in functionSet* **then**
    **return** false;
**else**
    **if** *ct.sender does not exist in f.Set* **then**
        **return** false;
    **else**
        **return** true;
    **end**
**end**

---

**Algorithm 2:** addUser($f$,*ct.sender*)

---

**Input:** $f$,ct.sender
**Output:** bool
**if** *f does not exist in functionSet* **then**
    **return** false;
**else**
    **if** *ct.sender has exist in f.Set* **then**
        **return** false;
    **else**
        f.set[ct.sender] $\Leftarrow$ true;
        **return** true;
    **end**
**end**

---

**Algorithm 3:** deleteUser($f$,*ct.sender*)

---

**Input:** $f$,ct.sender
**Output:** bool
**if** *f does not exist in functionSet* **then**
    **return** false;
**else**
    **if** *ct.sender has not exist in f.Set* **then**
        **return** false;
    **else**
        f.set[ct.sender] $\Leftarrow$ false;
        **return** true;
    **end**
**end**

---

4.2.2. Concrete Multi-Party Functional Signatures Scheme

This paper now describes the details of the multi-party functional signatures (MFS) scheme. This paper denotes the setup phase, the key generation phase, the signature phase, and the verification phase of the scheme as MFS.Setup, MFS.KeyGen, MFS.Sign, and MFS.Verify.

- *MFS.Setup:* The full node first selects the security parameter $\lambda$ and runs the Setup algorithm of the ECDSA signature to generate public parameters *param* = {E, $F_p$, G, P,

p, q, H}, then randomly selects $d \leftarrow Z_q^*$ as the master private key Msk and computes the master public key $d \cdot P$ as Mvk. The full node chooses n random numbers $d_i$ and computes $d_{n+1} = d - \sum_{i=1}^{n} d_i$, where satisfies $\sum_{i=1}^{n+1} d_i = d$.

- *MFS.KeyGen:* Each mining node $A_i$ initiates a request to the full node, and then the full node sends the $d_i$ to mining node $A_i$ by a secure channel. Each mining node $A_i$ selects $k_i \leftarrow Z_q^*$ randomly to send to the full node by Diffie–Hellman key exchange. The full node computes $k = \prod_{i=1}^{n} k_i$ and $k \cdot P = (r_x, r_y)$, where k is stored in the full node and $k \cdot P$ is sent to each mining node $A_i$. Let $\{v_1, v_2, \cdots, v_{n-1}\}$ be chosen randomly by the full node, and $v_n$ is computed by

$$v_n = k^{-1} - \sum_{i=1}^{n-1} v_i. \tag{1}$$

Each mining node sends H($Aid_i$) to the full node, where $Aid_i$ is the identification information of node $A_i$. The full node stores $\{d_i, v_i, H(Aid_i)\}_{i \in [1,n]}$ securely. The full node randomly selects $X_{ji} \leftarrow Z_q^*$ and computes $X_{ij} = d_i v_j + d_j v_i - X_{ji}$. Two nodes $A_i$ and $A_j$ receive $X_{ij}$ and $X_{ji}$, respectively, by Diffie–Hellman key exchange, where they satisfy

$$X_{ij} + X_{ji} = d_i v_j + d_j v_i. \tag{2}$$

Each mining node computes

$$X_i = d_i v_i + \sum_{j=1, j \neq i}^{n} X_{ij}. \tag{3}$$

The public key of mining node $A_i$ is $X_i P$. Each mining node $A_i$ gets $v_i$ by Diffie–Hellman key exchange as a part of the private key sk and invokes a smart contract to verify the authenticity of $f_i$. The full node signs for $f_i|vk_i$ by the Msk and SigGen algorithms of the ECDSA signature, the signature of $f_i|vk_i$ is noted as $\sigma_{vk_i}$. Creating the certificate $c_i = (f_i, vk_i, \sigma_{vk_i})$, the private key $sk_f^i$ of node $A_i$ is $sk_f^i = (X_i, c_i)$.

- *MFS.Sign:* To sign the message m, all the nodes start to get e = H(m) and compute

$$sig_i = e \cdot v_i + r_x \cdot X_i. \tag{4}$$

The signature generated by each mining node $A_i$ is $\sigma_i = (m_i^*, m, c_i, sig_i)$, where $m_i^* = f_i(m)$. The full node receives $sig_i$ from all nodes and calculates $sig_{n+1} = k^{-1} r_x d_{n+1}$ using the internally saved $d_{n+1}$. Finally, the output is the $Sig = \sum_{i=1}^{n+1} sig_i$.

- *MFS.Verify:* For overall transactions, the data of the transactions verifier can verify signature Sig by using the Verify algorithm of the ECDSA signature. If the data of transactions verifier needs to check the authenticity of the signature of node $A_i$, he can check that:

1. $m^* \stackrel{?}{=} f_i(m)$;
2. Whether $\sigma_{vk_i}$ is valid by the Verify algorithm of the ECDSA signature and Mvk.

- *Correctness:* The correctness of the signature for node $A_i$ is guaranteed by the ECDSA signature algorithm. The following equation determines the correctness of Sig:

$$
\begin{aligned}
Sig &= \sum_{i=1}^{n+1} sig_i \\
&= \sum_{i=1}^{n} (e \cdot v_i + r_x \cdot X_i) + sig_{n+1} \\
&= e \sum_{i=1}^{n} v_i + r_x \sum_{i=1}^{n} X_i + sig_{n+1} \\
&= ek^{-1} + r_x \sum_{i=1}^{n} (d_i v_i + \sum_{j=1, j\neq i}^{n} X_{ij}) + sig_{n+1} \\
&= ek^{-1} + r_x \sum_{i=1}^{n} d_i k^{-1} + sig_{n+1} \\
&= k^{-1}(e + r_x \sum_{i=1}^{n} d_i) + k^{-1} r_x d_{n+1} \\
&= k^{-1}(e + r_x d)
\end{aligned}
\tag{5}
$$

## 5. Security Analysis

This section uses the security model of [10] and give proof of the security of the multi-party functional signatures scheme and system model.

**Theorem 1.** *If the signature scheme ECDSA is existentially unforgeable under chosen message attacks, then the proposed multi-party functional signatures scheme as specified above satisfies the unforgeability requirement for functional signatures.*

**Proof of Theorem 1.** In the unforgeability game, one can define a PPT adversary noted as $\mathcal{A}_{MS}$, who can be allowed to make a query to the two random oracles denoted $\mathcal{O}_k$ and $\mathcal{O}_{sig}$. Then create a set [N] to represent all nodes. Let $\mathcal{S}$ be the subset of [N] and satisfy that the nodes in $\mathcal{S}$ are honest. The set of corrupted nodes is denoted as $S':=[N]\backslash S$. Assume that $\mathcal{Q}(\lambda)$ is the polynomial number and $\mathcal{A}_{MS}$ only can query $\mathcal{Q}(\lambda)$ times in the oracles $\mathcal{O}_k$ and $\mathcal{O}_{sig}$. This game can use $\mathcal{A}_{MS}$ as a subroutine to construct another adversary $\mathcal{B}_{MS}$ such that, if adversary $\mathcal{A}_{MS}$ wins in the unforgeability game for multi-party functional signatures with non-negligible probability, then it can show that $\mathcal{B}_{MS}$ can break the unforgeability game of the ECDSA signature scheme, which is assumed to be unforgeable against the chosen message attack.

For $\mathcal{A}_{MS}$ to win the unforgeability game of multi-party functional signatures, either it can generate a final signature Sig by corrupting a dishonest node or simply produce the signature $\sigma_i$ of honest node $A_i$, where $\sigma_i = (m_i^*, m, (f_i, vk_i, \sigma_{vk_i}), sig_i)$ such that:

- For each $A_i \in \mathcal{S}'$, $sig_i$ is a valid signature of m under the verification key $vk_i$.
- For each $A_i \in \mathcal{S}'$, $\sigma_{vk_i}$ is a valid signature of $f_i|vk_i$ under the master public key Mvk.
- $f_i(m) = m^*$ for all i.
- $\mathcal{A}_{MS}$ has not sent a query of form $(f_i', i)_{i\in\{i|(A_i\in\mathcal{S})\}}$ for all i to the oracle $\mathcal{O}_k$, and $m^*$ is in the range of the $f_i'$.
- $\mathcal{A}_{MS}$ has not sent a query of form $(f_i', i, m')_{i\in\{i|(A_i\in\mathcal{S})\}}$ for all i to the oracle $\mathcal{O}_{sig}$.

Let us discuss the first situation, where the $sig_i$ can be produced by $\mathcal{A}_{MS}$, where $i \in \{i|A_i \in \mathcal{S}\}$. It can assume that there are n-1 corrupted nodes of which the only uncorrupted node is noted as $A_{i^*}$. There are two types of forgery:

- *Type I forgery:* The tuples $(f_{i^*}, vk_{i^*})$ satisfy that $f_{i^*}|vk_{i^*}$ has not already been signed under the master key Mvk for queries from $\mathcal{A}_{MS}$ to the oracles $\mathcal{O}_k$ and $\mathcal{O}_{sig}$.
- *Type II forgery:* The tuples $(f_{i^*}, vk_{i^*})$ satisfy that $f_{i^*}|vk_{i^*}$ has been signed under the master key Mvk for queries from $\mathcal{A}_{MS}$ to the oracles $\mathcal{O}_k$ and $\mathcal{O}_{sig}$.

This game now can denote another adversary $\mathcal{B}_{MS}$ by the above. It first assumes that the all signatures with Mvk generated by oracles $\mathcal{O}_k$ and $\mathcal{O}_{sig}$ can be used as intermediate steps for responding to the $\mathcal{A}_{MS}$'s queries. In the unforgeability game for the ECDSA scheme, it defines that $\mathcal{B}_{MS}$ wins the unforgeability game only if he can output forged signatures successfully for a message that was not queried $\mathcal{O}_{Reg_{sig}}$, where $\mathcal{O}_{Reg_{sig}}$ is a oracle $\mathcal{B}_{MS}$ can use; $vk_{sig}$ is a verification key used by $\mathcal{B}_{MS}$.

As a challenger, $\mathcal{B}_{MS}$ needs to interact with $\mathcal{A}_{MS}$ in the unforgeability game for multi-party functional signatures; $\mathcal{B}_{MS}$ must simulate the oracles $\mathcal{O}_k$ and $\mathcal{O}_{sig}$ in order to interacting with $\mathcal{B}_{MS}$, and $\mathcal{B}_{MS}$ tosses a coin b to guess the type of forgery $\mathcal{A}_{MS}$ will generate.

*Case 1:* b = 1: $\mathcal{B}_{MS}$ guesses that $\mathcal{A}_{MS}$ will produce a *Type I* forgery:

During interacting with $\mathcal{A}_{MS}$, $\mathcal{B}_{MS}$ sets the $vk_{sig}$ as the master verification key in this simulation; $\mathcal{B}_{MS}$ generates and maintains a dictionary indexed by tuples $(f_i, i)$ to simulate the oracles $\mathcal{O}_k$ and $\mathcal{O}_{sig}$. Adversary $\mathcal{A}_{MS}$ will respond to the $\mathcal{B}_{MS}$'s queries as follows:

- $\mathcal{O}_k((f_i, i)_{i \neq i^*})$:
    - If there exists an entry for the tuple $(f_i, i)$ in the dictionary, then output the corresponding value $sk_f^{i^*}$.
    - Otherwise, $\mathcal{B}_{MS}$ randomly selects $d_{i^*} \leftarrow Z_q^*$ as $sk_{i^*}$, and computes $d_{i^*}P$ as $vk_{i^*}$; $\mathcal{B}_{MS}$ obtains $\sigma_{vk_{i^*}} \leftarrow \mathcal{O}_{Reg_{sig}}$ from his oracle, and return $sk_f^i = (sk_{i^*}, \sigma_{vk_i})$ to $\mathcal{A}_{MS}$. He also updates the dictionary by adding the entry $(f_i, i)$.

- $\mathcal{O}_{sig}((f, i, m)_{i \neq i^*})$:
    - If there exists an entry for the tuple $(f_i, i)$ in the dictionary, it has the $sk_f^i = (sk_i, \sigma_{vk_i})$. He can generate a signature $Sig \leftarrow \mathcal{O}_{Reg_{sig}}$. Because he knows all about the corrupted nodes, he can output $\sigma_i = (m_i^*, m, c_i, sig_i)$, where $c_i = (f_i, vk_i, \sigma_{vk_i})$ and $sig_i = Sig - \sum_{j=1, j \neq i^*}^n sig_j$.
    - Otherwise, $\mathcal{B}_{MS}$ randomly selects $d_{i^*} \leftarrow Z_q^*$ as $sk_{i^*}$, and computes $d_{i^*}P$ as $vk_{i^*}$; $\mathcal{B}_{MS}$ obtains $\sigma_{vk_{i^*}} \leftarrow \mathcal{O}_{Reg_{sig}}$ from his oracle, and updates the dictionary by adding the entry $(f_i, i)$. He then generates Sig by $\mathcal{O}_{Reg_{sig}}$, and outputs $\sigma_{i^*} = (m_{i^*}^*, c_{i^*}, sig_{i^*})$, where $c_{i^*} = (f_{i^*}, vk_{i^*}, \sigma_{vk_{i^*}})$ and $sig_{i^*} = Sig - \sum_{i=1, i \neq i^*}^n sig_i$.

Finally, $\mathcal{A}_{MS}$ outputs the signatures $\sigma_j$ and Sig, where $\sigma_j = (m_j^*, m, (f_j, vk_j, \sigma_{vk_j}), sig_j)$; $\mathcal{B}_{MS}$ then outputs the $(f_j | vk_j, \sigma_{vk_j})$ as a forged signature in the unforgeability game for ECDSA scheme.

*Case 2:* b = 0: $\mathcal{B}_{MS}$ guesses that $\mathcal{A}_{MS}$ will produce a *Type II forgery:*

Here, $\mathcal{B}_{MS}$ randomly selects $d \leftarrow Z_q^*$ as master private key Msk and computes dP as the master public key Mvk. Then $\mathcal{B}_{MS}$ generates $d_i$, $k \leftarrow Z_q^*$, $k \cdot p = (r_x, r_y)$, $v_i$ and $X_i$, where $d_i$ and $v_i$ satisfy that $\sum_{i=1}^n d_i = d$ and $\sum_{i=1}^n v_i = k^{-1}$, respectively; $\mathcal{B}_{MS}$ can compute $X_iP$ as $vk_i$ and sends the $vk_i$ of all nodes to $\mathcal{A}_{MS}$ along with the $X_i$ of the specified set $\mathcal{S}$ of corrupt nodes. In addition, $\mathcal{B}_{MS}$ guesses a value $i^*$ between 1 and $\mathcal{Q}(\lambda)$ as a special index of signing queries of multi-party functional signatures where the challenge verification key will be embedded. This game uses a variate denoted as *Numkeys* to indicate the numbers of signing keys already generated and initializes *Numkeys* = 0. Adversary $\mathcal{B}_{MS}$ maintains a dictionary indexed by tuples $(f_i, i)$ for responding to the queries of $\mathcal{A}_{MS}$; $\mathcal{B}_{MS}$ responses to the queries issued by $\mathcal{B}_{MS}$ as follows:

- $\mathcal{O}_k(f_i, i)$:
    - If there exists an entry for the tuple $(f_i, i)$ in the dictionary with the value CHA, abort.
    - If there exists an entry for the tuple $(f_i, i)$ in the dictionary with a value that is not CHA, output the key $sk_f^i$.

- Otherwise, $\mathcal{B}_{MS}$ randomly selects $d_i \leftarrow Z_q^*$ as $sk_i$, and computes $d_i P$ as $vk_i$; $\mathcal{B}_{MS}$ generates the $\sigma_{vk_i}$ by himself and returns $sk_f^i = (sk_i, \sigma_{vk_i})$ to $\mathcal{A}_{MS}$. He also updates the dictionary by adding the entry $(f_i, i)$.

- $\mathcal{O}_{sig}(f, i, m)$:
  - If there exists an entry for the tuple $(f_i, i)$ in the dictionary, $sk_f^i = (sk_i, \sigma_{vk_i})$. He generates a signature Sig by himself, and outputs $\sigma_i = (m_i^*, m, c_i, sig_i)$, where $c_i = (f_i, vk_i, \sigma_{vk_i})$ and $sig_i = Sig - \sum_{j=1, j \neq i^*}^n sig_j$.
  - If there is no the tuple $(f_i, i)$ in the dictionary, and $Numkeys \neq i^*$, $\mathcal{B}_{MS}$ generates a new key pair by randomly choosing the $d_i \leftarrow Z_q^*$ as $sk_i$ and computing $d_i P$ as $vk_i$; $\mathcal{B}_{MS}$ signs $f_i | vk_i$ to generate $\sigma_{vk_i}$ by using Msk, and sets tuple $(f_i, i)$ in his dictionary to $sk_f^i$. He then generates the signature Sig on message m by using Msk, outputs $\sigma_i = (m_i^*, m, c_i, sig_i)$, where $c_i = (f_i, vk_i, \sigma_{vk_i})$ and $sig_i = Sig - \sum_{j=1, j \neq i^*}^n sig_j$, and sets $Numkeys = Numkeys + 1$.
  - If there is no tuple $(f_i, i)$ in the dictionary and $Numkeys = i^*$, or if the tuple $(f_i, i)$ in the dictionary is set to CHA, $\mathcal{B}_{MS}$ then generates the signature of m under $Mvk'$ by oracle $\mathcal{O}_{Reg_{sig}}$, $Sig \leftarrow \mathcal{O}_{Reg_{sig}}$, computes $\sigma_{vk_i}$ by using Msk, and outputs $\sigma_i = (m_i^*, m, c_i, sig_i)$, where $c_i = (f_i, vk_i, \sigma_{vk_i})$ and $sig_i = Sig - \sum_{j=1, j \neq i^*}^n sig_j$. If there is no tuple $(f_i, i)$ in the dictionary, $\mathcal{B}_{MS}$ sets it to CHA. Then $Numkeys = Numkeys + 1$.

If $\mathcal{B}_{MS}$ does not abort, $\mathcal{A}_{MS}$ finally outputs a signature Sig on m; $\mathcal{B}_{MS}$ outputs $(m.Sig)$ as the forged signature in this unforgeability game for the ECDSA signature scheme.

It can indicate that, if $\mathcal{A}_{MS}$ breaks the multi-party functional signatures scheme with non-negligible probability, then $\mathcal{B}_{MS}$ can break ECDSA scheme with non-negligible probability. In this unforgeability game, $\mathcal{A}_{MS}$ has two different types of forgery.

For *Type I forgery*, this game can know that this forgery has a signature on the new message $f_i | vk_i$ that was never signed under the master public key Mvk by the oracle. In *Type I forgery*, this game assumes that the probability of breaking the multi-party functional signatures scheme is Pr[I], and it is obvious that the probability of $\mathcal{B}_{MS}$ breaking the ECDSA scheme is greater than or equal to $\frac{1}{2}$Pr[I].

For *Type II forgery*, when $\mathcal{A}_{MS}$ generates a *Type II forgery*, the corresponding $f_i | vk_i$ should have been signed under the master public Mvk during the queries of $\mathcal{A}_{MS}$ to the oracles $\mathcal{O}_k$ and $\mathcal{O}_{sig}$. It shows that the tuple $(f_i, vk_i)$ cannot be queried by $\mathcal{A}_{MS}$ to $\mathcal{O}_k$ because the signature generated under the signing key is not a valid forgery in the multi-party functional signatures scheme if it is subsequently the signing key for the response. Therefore, the oracle $\mathcal{O}_{sig}$ already has been issued the tuple $(f_i, i, m)$. It can note that if $\mathcal{A}_{MS}$ does abort, it must be that he embedded his challenge in a query $i^*$ with the form $\mathcal{O}_{sig}(f, i, m)$. In *Type II forgery*, it assumes that the probability of breaking the multi-party functional signatures scheme is Pr[II], and the probability of $\mathcal{B}_{MS}$ breaking the ECDSA scheme is greater than or equal to $\frac{1}{2\mathcal{Q}(\lambda)}$Pr[II].

Thus, if $\mathcal{A}_{MS}$ can generate a forgery in the multi-party functional signatures scheme with non-negligible probability Pr[I] + Pr[II], then $\mathcal{B}_{MS}$ can break the ECDSA scheme with non-unforgeability probability $\frac{1}{2\mathcal{Q}(\lambda)}$(Pr[II] + Pr[II]). However, the ECDSA scheme is existentially unforgeable under chosen message attacks, and the probability of the $\mathcal{B}_{MS}$ successfully breaking the ECDSA scheme is negligible. It can show that $\mathcal{A}_{MS}$ generates a forgery in the multi-party functional signatures scheme with negligible probability. $\square$

**Theorem 2.** *The multi-party functional signatures in this paper have collision resistance.*

**Proof of Theorem 2.** In the multi-party functional signatures scheme, it needs to be assumed that the full node is secure. The master private key of the scheme is $d = \sum_{i=1}^{n+1} d_i$; therefore, even if n − 1 nodes are corrupted, the adversary cannot recover d by corrupted nodes. Furthermore, the nodes have $X_i = d_i v_i + \sum_{j=1, j \neq i}^n X_{ij}$, even if n-1 corrupted nodes

provide $X_i$, the adversary cannot compute $X_{i^*}$ that the honest node has, so it cannot get $sig_{i^*}$. The above illustrates this scheme resists n-1 corrupted nodes collusion attacks. □

## 6. Results

This section analyzes the computation complexity and implements the proposed system model on a personal computer.

### 6.1. Complexity Analysis

As shown in Table 2, this paper gives a theoretical analysis of the proposed scheme. Some notations of time complexity analysis are given as follows.

**Table 2.** Theoretical cost analysis.

|  | Mining Node | Full Node |
| --- | --- | --- |
| Time costs | $T_{A_i} = T_h + T_{Gmul} + 3T_{Zmul} + (n+1)T_{Zadd} + T_{sc} + nT_{dh}$ | $T_S = nT_h + 2T_{Gmul} + (n^2 + 2n - 1)T_{Zmul} + \frac{n^2+3n}{2}T_{Zadd} + (n+1)T_{inv} + nT_{dh}$ |

- $T_h$: Time costs to run one hash function;
- $T_{Gmul}$: Time costs of running one multiplication operation in additive group G;
- $T_{Zmul}$: Time costs of running one multiplication operation in field $Z_q^*$;
- $T_{Zadd}$: Time costs of running one addition operation in field $Z_q^*$;
- $T_f$: Time costs of running one $f(m)$;
- $T_{com}$: Time costs for full node to communicate with nodes;
- $T_{inv}$: Time costs of running one extended Euclidean algorithm;
- $T_{sc}$: Time costs of running one smart contract;
- $T_{dh}$: Time costs of running one Diffie–Hellman key exchange.

In the proposed scheme, each mining node must perform one hash function, three multiplications in the field $Z_q^*$, $n+1$ additions in the field $Z_q^*$, one multiplication in G, one $f(m)$, and $n$ Diffie–Hellman key exchanges. Each mining node needs to invoke one smart contract to use function $f$. Excluding communication time, each mining node will cost

$$T_{A_i} = T_h + T_{Gmul} + 3T_{Zmul} + (n+1)T_{Zadd} + T_{sc} + nT_{dh}. \tag{6}$$

For the full node, it needs to run some random number generation algorithm and ordinary addition and multiplication operations several times (the time cost of these steps is negligible). In addition, it runs the n hash function, two multiplications in G, $n(n-1) + 2n + (n-1)$ multiplications in the field $Z_q^*$, $n(n-1)/2 + n + (n-1) + 1$ additions in field $Z_q^*$, $n+1$ extended Euclidean algorithms, and n Diffie–Hellman key exchanges. Therefore, the full node requires

$$T_S = nT_h + 2T_{Gmul} + (n^2 + 2n - 1)T_{Zmul}$$
$$+ \frac{n^2 + 3n}{2}T_{Zadd} + (n+1)T_{inv} + nT_{dh}. \tag{7}$$

For communication between the full node and mining node, it requires $(n^2 + 2n)T_{com}$.

The scheme presented in this paper implements fine-grained authentication for certain transactions, which is highly significant. Additionally, for multi-party signature schemes, it is essential that they satisfy resistance to collusion attacks and can be proven secure. As is shown in Table 3, the proposed scheme has certain advantages compared to the previous scheme. The proposed scheme ensures that every part of the transaction is validated, reflecting the fine-grained nature of the scheme. Furthermore, the scheme is adapted to run on multiple nodes of the blockchain, and it is provably secure. For the collusion attack, the proposed scheme can effectively resist such attacks. Even if $n - 1$ nodes are corrupted, the

adversary cannot forge the signature generated by this scheme. The significant advantage of this paper's scheme over previous schemes is that it can achieve fine-grained authentication of different parts of the transaction and resist collusion attacks.

**Table 3.** Properties and disadvantages of the schemes.

| Scheme | Blockchain | Provable Security | Against Collusion Attack | Fine-Grained |
|---|---|---|---|---|
| [24] | no | yes | no | no |
| [33] | yes | uncertain | no | no |
| [38] | yes | yes | no | no |
| [39] | yes | yes | yes | no |
| [40] | yes | yes | yes | no |
| our | yes | yes | yes | yes |

*6.2. Implement*

This paper used a personal computer (RedmiBook with AMD CPU Ryzen 5 5600H with Radeon Graphics @ 3.30 GHz with 16.0 GB RAM and Windows 10 Home OS) implementing the proposed scheme in Python 3.9.6. This paper implements the scheme with different numbers of nodes.

This paper chose the secp256k1 curve that is used in Bitcoin for the simulation experiment. This paper implemented the experiments with different numbers of nodes. Each execution was run 1000 times, and this paper computed the average as the final result. The results of the experiment are shown in Figure 4. It can see that the running time of the scheme slows down as the number of nodes increases: about 343 ms for three nodes, about 552 ms for six nodes, and about 791 ms for nine nodes. Regarding time consumption, the scheme in this paper can be applied to the private blockchain.

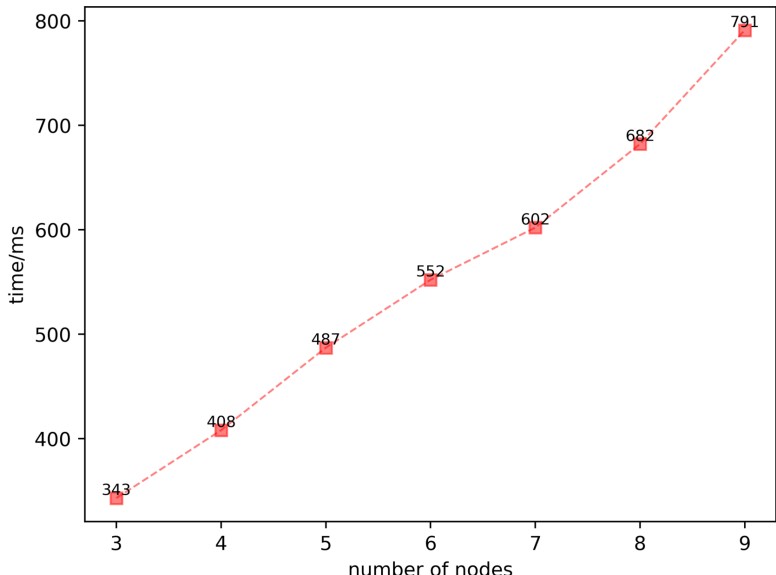

**Figure 4.** Running times for different numbers of nodes.

This paper deployed smart contracts using Solidity language and Ropsten to test the network. The compiler of Solidity language is Remix IDE, and its version is 0.7.4.We. This paper used MetaMask to manage private blockchain accounts. In the experiment, the gasPrice was set to 1.49 *Gwei*, where 1 *Gwei* = $10^9$ *wei* = $10^{-9}$ *ether*. This paper defines $|functionSet|$ as the length of set functionSet. As shown in Tables 4 and 5 and Figure 5, some costs of smart contracts were tested through this experiment.

**Table 4.** Cost of deploying smart contract (*gasprice* = 1.49 *Gwei* and 1 *ether*= $1026).

| Length of FunctionSet | Gas Used | USD ($) |
| --- | --- | --- |
| 5 | 948,328 | 0.973 |
| 10 | 1,115,115 | 1.144 |
| 15 | 1,281,915 | 1.315 |
| 20 | 1,448,660 | 1.486 |

**Table 5.** Cost of smart contract under different lengths of functionSet (*gasprice* = 1.49 *Gwei* and 1 *ether* = $1026).

| Length of FunctionSet | Function | Gas Used | USD ($) |
| --- | --- | --- | --- |
| 5 | invokeF | 64,445 | 0.066 |
| | addUser | 94,981 | 0.097 |
| | deleteUser | 55,675 | 0.057 |
| 10 | invokeF | 83,965 | 0.086 |
| | addUser | 114,502 | 0.117 |
| | deleteUser | 75,195 | 0.077 |
| 15 | invokeF | 103,486 | 0.106 |
| | addUser | 134,023 | 0.138 |
| | deleteUser | 94,716 | 0.097 |
| 20 | invokeF | 123,007 | 0.126 |
| | addUser | 153,543 | 0.158 |
| | deleteUser | 114,237 | 0.117 |

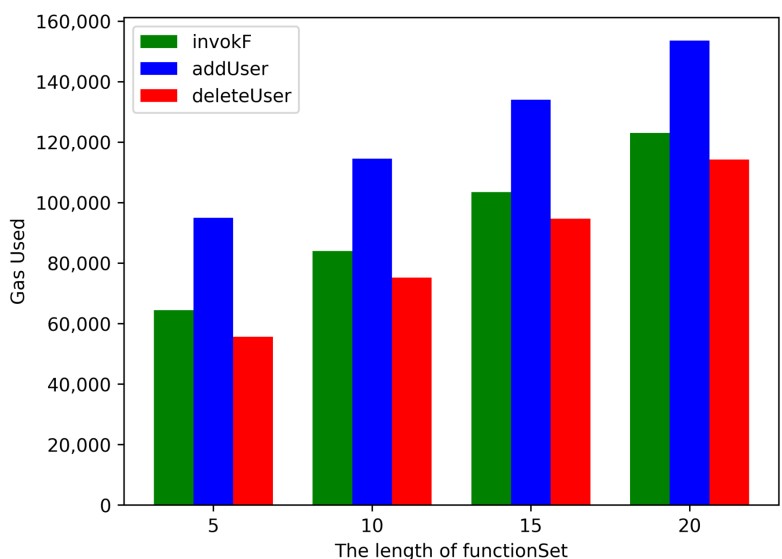

**Figure 5.** Smartcontract costs under different lengths of functionSet.

Costs of smart contract deployment and algorithms increase as the functionSet length increases. In this experiment, the cost of deploying smart contract for $|functionSet| = 5$ is $0.973. When the length of functionSet is 10, 15, and 20, the cost of deploying smart contract is $1.144, $1.315, and $1.486. As shown in Table 5 and Figure 5, when $|functionSet| = 5$, gasPrice of invokeF, addUser, and deleteUser is 64,445 *Gwei*, 94,981 *Gwei* and 55,675 *Gwei*, respectively. Obviously, with the increase of the $|functionSet|$, the gasPrice required to invoke the smart contract also increases. GasPrice of addUser is always more than invokeF and deleteUser. All gasPrice consumption is reasonable, which indicates that binding function $f$ of the function signature to a private smart contract is valid.

*6.3. The Limitations of the Study*

This paper is preconfigured to use the function signature of function $f$ in the smart contract, which may limit the operability of this scheme. Additionally, this paper is directly combined with ECDSA signatures and does not apply to other elliptic curve signatures.

## 7. Conclusions

Although some researchers have proposed various multi-party signatures to tackle the issue of verifying private blockchain transactions by multiple parties, the previous construct still needs to address authenticating different parts of a transaction. This paper presents a novel approach that combines multi-party ECDSA signatures with functional signatures to enable fine-grained authentication of different parts of the entire transaction. In addition, the function $f$ of the functional signature is safeguarded and implemented using a smart contract on a private blockchain. The research presented in this paper is motivated by the fact that previous multi-party signature schemes designed for blockchains cannot authenticate the entire transaction at a fine-grained level. Furthermore, when the final signature fails to pass the authentication, the whole transaction will be rejected, which greatly wastes time for the partly correct transaction to pass the authentication. Experimental results show that the proposed scheme is feasible to be applied to the blockchain.

Exploring how to design an efficient and multi-party signature with an automatic fine-grained division of blockchain transactions is an important research direction for the future. The multi-party functional signatures proposed in this paper will also find different applications outside of the private blockchain.

**Author Contributions:** Conceptualization, Y.Z. and Q.Z.; methodology, Y.Z.; software, Y.Z, K.W. and M.C.; validation, Y.Z., Q.Z. and Z.Z.; formal analysis, Y.Z.; investigation, Y.Z.; resources, Y.Z.; data curation, Y.Z.; writing—original draft preparation, Y.Z.; writing—review and editing, Y.Z. and Q.Z.; visualization, Y.Z.; supervision, Q.Z.; project administration, Q.Z.; funding acquisition, Q.Z. All authors have read and agreed to the published version of the manuscript.

**Funding:** This research was funded by The National Key R&D Program of China (grant number 2021YFA1000600) and National Natural Science Foundation of China (grant number 12171114).

**Institutional Review Board Statement:** Not applicable.

**Informed Consent Statement:** Not applicable.

**Data Availability Statement:** Not applicable.

**Conflicts of Interest:** The authors declare no conflict of interest.

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
