# Peer review of "A Multi-Party Functional Signatures Scheme for Private Blockchain"

_cryptography, doi:10.3390/cryptography7020021_

Round 1

Reviewer 1 Report

Paper title:  A Multi-Party Functional Signatures Scheme For Private

Blockchain

 There are some points that need to be further clarified:

1-       The motivation for the study should be further emphasized, particularly; the main advantages of the results in the paper comparing with others should be clearly demonstrated. 

2-       The limitations of the study are better suited for the discussion in a separate sub-section after the discussion on results.

3-       The example section needs to be further expanded and include some remarks to show the effectiveness and efficiency of the proposed method, compared with others. 

4-       Some remarks on the main results would be necessary and helpful. 

5-       The literature review should be extended.

Recommendation: According to all these issues,

Decision: major revisions

Author Response

Dear reviewer, 

Thank you for the valuable suggestions you have provided. I have made the requested changes, please see the uploaded PDF file for details.

Reviewer 2 Report

I recommend a major revision based on the below points. Please, add a point-to-point response to each comment in the revision:

-          I am not convinced about the novelty of the manuscript. The novelty of the paper needs to be justified and clearly defined. It includes a clear difference between the available literature and previous works. The authors are asked to provide the limitations of the previous correlated works and then link those limitations to the current ideas and contributions of the current work.

-          What is ECDSA in the abstract?

-          Add more keywords.

-          Explain each step of Figure 2.

-          The abstract is not technical and needs to highlight the research gap clearly.

-          The abstract also missed statistical information about the results.

-          The structure of the paper is vague. The paper needs to be restructured.

-          Do not add heading over heading. Instead, add a few lines related to the detail of a particular section before starting a sub-section.

-          Please avoid using the words "you," "we," or "our" in the manuscript. Please, consider using phrases like "in this study/paper/Proposed/method" or another appropriate phrasing. This applies to the entire manuscript.

-          Proofread the manuscript from a native English speaker. There are many typos and grammar mistakes. 

-          Generally, the entrance to the subject should be done more clearly and briefly. Consequently, please rewrite the introduction section and the related works section accordingly.

-          The heading should be literature review/related work.

-          At the end of the Introduction section, add the contributions clearly. See this paper for reference and citation ‘A novel deep learning-based approach for malware detection.' The mentioned contributions are not clear.

-          Related work/background/literature review should have a threat to a validity section. Add the ‘threat to a validity’ section at the start of the background section. In that section, state the search strings and databases explored to find the related work. See the below papers for references and citations 'Performance comparison and current challenges of using machine learning techniques in cybersecurity' and 'A Survey on Machine Learning Techniques for Cyber Security in the Last Decade'.

-          Summarise the literature in the form of a table.

-          Add a discussion related to adversarial attacks. See this paper: A Novel Method for Improving the Robustness of Deep Learning-based Malware Detectors against Adversarial Attacks

-          The literature needs to be subdivided into multiple sub-sections.

-          Add the below papers to your literature: A Review on Security Challenges in Internet of Things (IoT), A Review of Time-Series Anomaly Detection Techniques: A Step to Future Perspectives, What is Core and What Future Holds for Blockchain Technologies and Cryptocurrencies: A Bibliometric Analysis, A Review of Content-Based and Context-Based Recommendation Systems

-          Comparison with the state-of-the-art is missed. You need to compare your method with the ground truth. 

-          It needs to add the reasons why these metrics are used for comparison. 

-          Add the discussion related to the time complexity factor of AI models. See this paper for reference and citation 'Cyber Threat Detection Using Machine Learning Techniques: A Performance Evaluation Perspective'.

-          The authors need to perform any statistical test to validate the significance of their proposed method.

Overall, the paper has many inconsistencies, and the contributions are unclear. The results are not compared with the ground truth properly. Limitations are not provided in their current approach. Future directions are not clearly stated.

I am looking forward to seeing your revised version. 

All the best. 

Author Response

(The authors gave the same response as above.)

Reviewer 3 Report

The abstract must be reviewed in the sense that the authors must present the essence of the results obtained. Also, the obtained results must be corroborated with the CONCLUSIONS section. It is very important that the CONCLUSIONS section be extended and present the results in an objective manner. Unfortunately, the authors talk about them in the conclusions and not about the conclusions of the study. I strongly recommend rewriting the conclusions.

Author Response

Dear reviewer,

Thank you for your advice. I have fully rewritten the abstract and conclusions as you suggested. The specific rewritten conclusions are as follows:

“While some researchers have proposed various multi-party signatures to tackle the issue of verifying private blockchain transactions by multiple parties, the previous construct still needs to address authenticating different parts of a transaction. This paper presents a novel approach that combines multi-party ECDSA signatures with functional signatures to enable fine-grained authentication of different parts of the entire transaction. In addition, the function f of the functional signature is safeguarded and implemented using a smart contract on a private blockchain. The research presented in this paper is motivated by the fact that previous multi-party signature schemes designed for blockchains cannot authenticate the entire transaction at a fine-grained level. Furthermore, when the final signature fails to pass the authentication, the whole transaction will be rejected, which greatly wastes time for the partly correct transaction to pass the authentication.Experimental results show that the proposed scheme is feasible to be applied to the blockchain.

Exploring how to design an efficient and multi-party signature with an automatic fine-grained division of blockchain transactions is an important research direction for the future. The multi-party functional signatures proposed in this paper will also find their different applications outside of the private blockchain.”

Round 2

Reviewer 1 Report

The authors have addressed all my comments for this paper and answered the technical questions I have for this method.

Reviewer 2 Report

The authors have well addressed my comments. 

Congratulations. 

Reviewer 3 Report

The authors kept close the necessary advice and corrections. The quality of the manuscript is greatly improved.